# Spatial Exposure Responses of Malaria Vectors to *Eucalyptus grandis* (W. Hill ex Maiden) and *Cymbopogon citratus* (DC.) Stapf Essential Oils

**DOI:** 10.3390/biology14121768

**Published:** 2025-12-11

**Authors:** Martha A. Kaddumukasa, Norah M. Mutekanga, Faisal Kula, Charles Batume, Agapitus B. Kato

**Affiliations:** 1Department of Biological Sciences, Faculty of Science, Kyambogo University, Kampala P.O. Box 1, Uganda; nmutekanga@kyu.ac.ug (N.M.M.); faizalmartins87@gmail.com (F.K.); 2Department of Entomology, Uganda Virus Research Institute, Nakiwogo, Entebbe P.O. Box 49, Uganda; cbatume@uvri.go.ug (C.B.); akato@uvri.go.ug (A.B.K.)

**Keywords:** *Anopheles gambiae*, *Cymbopogon citratus* EO, *Eucalyptus grandis* EO, gas chromatography–mass spectrophotometry, spatial activity index

## Abstract

Mosquito-borne diseases affect millions of people around the world, including Uganda. People therefore need to protect themselves from mosquito bites to stop the spread of disease. One way of keeping mosquitoes at bay is through the use of natural aromatic plant compounds as repellents. These plants release compounds (essential oils) as vapors into the air that keep mosquitoes away, creating a “mosquito-free zone”. In this research, the repelling power of Eucalyptus and lemon grass leaves against mosquitoes and their composition were studied. Leaves from the Nwoya district community, Uganda, were identified and essential oils were obtained by distillation. The oils were tested on Anopheles mosquitoes in an olfactometer in a laboratory. The testing showed that eucalyptus and lemon grass do indeed repel mosquitoes, with eucalyptus showing more effectiveness than lemon grass. A number of compounds were discovered from the gas chromatography–mass spectrophotometric analysis, many of which were similar to other findings but varied in percentage composition. Incorporating Eucalyptus and lemon grass essential oils with spatial activity into control programs can complement existing tools used, such as mosquito nets, offer community-based, low-cost protection, and provide alternatives in the midst of rising parasite, drug, and insecticide resistance.

## 1. Introduction

Personal protection against mosquito bites is an additional layer of defense in malaria-endemic areas and especially against other mosquito-borne diseases [1,2,3]. This protection is afforded by the application of substances either from chemicals or plants (compounds, extracts, or essential oils) to deter mosquitoes from biting [2,3]. These substances are called mosquito repellents and are sometimes referred to as volatile compounds [4,5,6]. Mosquito repellents work by keeping mosquitoes away from human skin and their source, avoiding contact, bites, and disease [3,4]. Repellents, whether artificial or derived from natural sources, are substances applied to the skin or clothing to stop contact with insects [1,2,3,4]. Mosquito repellent substances are either many plant extracts [7,8,9,10,11,12], plant compounds [8,9], or essential oils [2,3,9,13]. Artificial repellents, typically containing DEET or picaridin, have been shown to offer effective protection [1,11,14]. Various chemical and natural plant repellents, such as *Azaradirachta indica* (Neem tree), *Citrus nardus* (Citronella), *Cymbopogon citratus* (Lemon grass and certain plant extracts), and *Eucalyptus grandis* (Eucalyptus) have been explored for their potential mosquito-repelling properties, spatial activity, and have shown beneficial protection [2,3,15,16].

Among the many studies performed are those that have demonstrated the insecticidal [17,18], repellent [19,20], larvicidal [21,22,23,24,25], and adulticidal [26,27] effects of various essential oils [26,27,28,29,30], highlighting their potential as eco-friendly mosquito control agents. The composition of many have also been explored [4,15,30]. However, these studies often differ in the materials and methods used [20,26], the type of formulations examined [5,6,17,18,19], and mosquito species tested [20,26,27], leading to variable and sometimes inconclusive results. Many investigations have focused on short-term repellency [23,26,27], and fewer have assessed the spatial efficacy, longevity, or scalability of essential oils in real-world applications [31]. The use of repellent substances in various forms can effectively prevent mosquito bites to human hosts [5,6]. These EOs found in plant compounds [5,6] are used as aromatic plant compound extracts, plant alkaloids, and EOs to control mosquitoes [1,2,3,4]. EOs are manufactured in the plastids and cytoplasm of plant cells, 2-C-methyl-erythritol 4-phosphate (MEP), and mevalonic acid pathways, respectively [15,16,17]. They are made up of various oxygenated hydrocarbons, phenylpropenes, and volatile hydrocarbon molecules (terpenes and sesquiterpenes) [16,17,31]. EOs greatly influence mosquito activity in a number of ways, such as by disrupting mosquito-seeking behavior when used as indoor spraying products or applied on human skin [11], plant extracts [5,6,12], compounds [13], and EOs [9,12,14].

EOs influence the neurological system of insects by obstructing GABA-gated chloride channels, octopamine receptors, and the enzyme acetylcholinesterase [18,19]. Monoterpenes constitute above 90% of the essential oils that can cause acetylcholinesterase enzyme activity inhibition in insects [20,32,33,34]. EOs also target the octopaminergic system of insects by blocking octopamine receptors causing acute behavioral effects on insects [35,36,37]. This is accomplished by blocking the rise in cyclic AMP (Adenosine Mono phosphate) levels as a result of the binding of octopamine receptors with a combination of EOs, including eugenol, y-terpineol, and cinnamic alcohol. Monoterpenes found in EOs, such as citronellal, linalool, and estragole, can also inhibit GABA-gated chloride channels [38,39,40]. They do this by first binding to receptor sites, which increases the influx of chloride anions into neurons and causes hyper excitation of the central nervous system, convulsions, and ultimately insect death [41,42].

Most sesquiterpenes and monoterpenes of EOs are known to contain repellent activities [25,26]. For example, ρ-cymene, α-pinene, eugenol, γ pinene, citronellol, citronellal, and camphor, among monoterpenes, are responsible for the repelling of mosquitoes [27,28,35]. On the other hand, sesquiterpenes like α-bisabolol, guaiol, nootkatone, α-cadinol, β-caryophellene, and germacrene have been identified as repellent representatives [13,31]. β-caryophellene expresses high repellent activity against *Aedes* mosquitoes [27,41]. EOs of *Artemisia absinthium* have demonstrated harmful effects on populations of *Anopheles*, *Aedes*, and *Culex* mosquito larvae [29]. EOs have also been investigated to cause toxicity and repellency at various developmental stages against adult *Anopheles mosquitoes* [20,24,41,42,43,44,45]. Ovicidal and larvicidal activities have been observed against *An. gambiae* and *Ae. aegypti* from EOs got from *Lippa multiflora*, *Cymbogon proximus*, and *Ocimum canum* [43,46,47]. Combinations of EOs tested on *Anopheles gambiae* mosquitoes showed that blended EOs yielded stronger repellency than individual EOs [48,49]. Consequently, the major constituents of EOs work together to provide increased insect toxicity, repellant, or larvicidal effects [17,24]. Because they target distinct areas or have different modes of action, EOs with a combination of active components have been shown to diminish resistance in mosquito populations [30,34,50]. Combinations of EOs tested on *Anopheles gambiae* mosquitoes showed that blended oils yielded stronger repellency than individual EOs. Consequently, the major constituents of EOs work together to provide increased insect toxicity and repellant and larvicidal effects [30,32,34] because they target distinct areas or have different modes. A number of chemical constituents of EOs have been explored and these are responsible for their antioxidative, antimicrobial, and pharmaceutical effects, as well as repellent and insecticidal effects [23,24,25,26,27,30,39].

However, the overdependence and inappropriate application of synthetic pesticides has led to environmental degradation, emergence of resistant strains, and toxicity in mammals despite their effectiveness in controlling vectors. The use of chemically derived mosquito repellents has raised concerns of insecticide resistance, environmental persistence, and potential adverse health effects. As a result, there is growing scientific interest in plant-derived alternatives, particularly EOs, due to their bioactive properties, biodegradability, and relative safety [21]. Despite ongoing efforts to control the disease through various means, the persistence of malaria cases indicates a need for innovative and sustainable vector control strategies. One promising avenue for mosquito control involves exploring the repellent activity of essential oils derived from aromatic plants. This research study addresses these gaps by examining the spatial activity responses of the EOs of *Eucalyptus grandis* and *Cymbopogon citratus* against *Anopheles gambiae* mosquitoes to inform the practicability of their use in integrated vector management strategies.

## 2. Materials and Methods Study Area

Nwoya District is situated in the northern Ugandan Acholi sub region at 02°38′ N, 32°00′ E. The district is neighbored by six districts: Oyam to the east; Kiryandongo in the southeast; Gulu in the northeast; Masindi to the south; Buliisa to the southwest; and Amuru to the north. The area experiences both wet and dry seasons due to its elevated location at 3220 feet above sea level. The wet season occurs from March to November and is characterized by warm and humid conditions, whereas the dry season occurs from December to February and is usually hot. The months of May, June, August, and October receive the highest amount of rainfall annually. Annual variation in temperature is between 18 degrees Celsius and 36 degrees Celsius [48]. Administratively, Nwoya district has four counties with eleven sub counties (Figure 1). With 133,500 residents and a population density of 37 people per square kilometer, the district is primarily rural, with an estimated 10% annual population growth rate [49]. There are three health center IIIs, fourteen health center IIs, and one main hospital in the district [49].

### 2.1. Study Design

The study employed a cross-sectional study design supplemented by experimental procedures [51].

### 2.2. Aromatic Plant Essential Oil Extraction

Plants were collected from various parts of Nwoya district, Uganda. EOs from the fresh leaves of the gathered plants were extracted using the steam distillation method [52]. Fresh leaves were chopped using a knife and placed in a stainless-steel distiller; the plant materials with known mass underwent steam distillation. To prevent direct contact with the plant, materials were packed in a sieved compartment inside the vessel, with two liters of distilled water at the bottom of the stainless-steel vessel. During the steam distillation process, volatile compounds were released from the packed plant material while being cooled by the steam. For three hours of the process, the distillate containing water and plant volatiles was gathered in a separating funnel. By the process of decanting, essential oils were extracted from the top layer of the mixture into a separating funnel. After the extraction process, the residual distillate was sent to liquid–liquid extraction (LLE), it was dried by adding anhydrous magnesium sulfate to 70 mL × 3 HPLC grade n-hexane. The extracted hexane layers were pooled, filtered, and the solvent evaporated under reduced pressure using a rotary evaporator at 25 °C. The extracted EOs by LLE were obtained by decanting, weighed and pooled together. The mass of EO was divided by the initial mass of plant material and multiplied by 100 to obtain the percentage yield of essential oil obtained. All extracted EOs were kept in glass vials at −20 °C.

### 2.3. Mosquito Culture

For all experiments conducted in this study, female *Anopheles gambiae* Kisumu strains from Uganda Virus Research Institute (UVRI) insectaries were used. Batches of approximately 500 eggs were hatched in 33 × 51 × 5 cm pans containing three liters of tap water. Fish flakes were fed to the larvae ad libitum. Pupae were sorted into 200 mL plastic cups and transferred to BugDorm-1 Insect Rearing Cages (30 × 30 × 30 cm, Bug dorm Company, Taichung, Taiwan). Cotton wool containing 10% sucrose solution was put on the top of each cage for feeding adult mosquitoes and changed weekly. Cages were kept inside an insectary room that was maintained at room temperature (average 27 °C) and 70–80% humidity with a light/dark cycle of 12/12, respectively.

### 2.4. Dilution of Test Compounds

In a 14 mL falcon tube, dilutions of the *E. grandis* and *C. citratus* test compounds were prepared using a micropipette for a range of concentrations ranging from 0 to 100% and diluted at intervals of 10% using pure olive oil as a diluent. The tubes were vortexed for sixty seconds before use.

### 2.5. Spatial Repellency Assay

In the spatial repellency test, the mosquito is elicited to move away from the treated surface without making physical contact [53]. To assemble a spatial repellency assay system, the treatment and control chambers were connected to opposite sides of the clear cylinder using the linking section [54]. With the butterfly valves in both linking sections closed, 25 mosquitoes were transferred into the clear cylinder. After 30 s of acclimatization, the butterfly valves were simultaneously opened and closed again after 10 min. The number of mosquitoes in the treatment and control chambers, as well as the clear cylinder were recorded. A total of three replications (utilizing 25 female mosquitoes in total) were conducted for each treatment (*n* = 3) and for each concentration (*n* = 3).

### 2.6. Repellence Testing Procedure

Mosquito repellence of the test compounds was tested using a standard Y-tube olfactometer [51,55] (Figure 2) in accordance with WHO guidelines [56]. Mosquitoes were starved for three to four hours prior to performing each experiment. The olfactometer was cleaned with 70% ethanol before and between experiments. Repellence assays were performed between 8:00 a.m. and 6:00 p.m. with presence of natural light. In the olfactometer, the control center was exposed to pure olive oil while the treatment side was exposed to different EO test compounds.

### 2.7. Data Analysis

Spatial repellency was expressed as the proportion of mosquitoes prevented from entering the treatment space in relation to all mosquitoes moving within the system and was calculated from a ‘spatial activity index’ using the equation below. $$SAI=\left[\frac{Nc-Nt}{Nc+Nt)}\right]\times (\frac{Nm}{N})$$

The spatial activity index varies from −1 to 1: zero indicates no response; −1 indicates that all mosquitoes moved into the treatment chamber, resulting in an attractant response, and 1 indicates that all the mosquitoes moved into the control chamber (away from the treatment source), resulting in a spatial repellent response. If no movement is recorded within the system (i.e., *Nt* = 0, *Nc* = 0), the test is valid but the spatial activity index is 0. The spatial activity index was calculated for each replicate, and the mean index for each activity index dosage was analyzed by Probit-plane regression analysis, from which the ED_50_ and ED_90_ and the corresponding 95% confidence limits were estimated. The number of replicates, the total number of mosquitoes, and the mean spatial activity index (±standard error) for each activity index dosage and negative control were recorded.

### 2.8. Gas Chromatography–Mass Spectrometry (GC-MS) Analysis

For the general identification of each plant composition, the extracts of individual plants were analyzed by GC-MS of Agilent Technologies (Santa Clara, CA, USA) model (Intuvo 9000 GC connected to 19091S-433UI-INT MS) with a HP-5MS UI column with 30 m length, 250 μm dimensions, and 0.25 μm film thickness. Helium served as the carrier gas and the flow rate was set at 3 mL/min. This process determined the names, molecular weights, and molecular formulae of the components present in the extracts. To ensure validity of the results, the instrument was tuned using perfluorotributylamine (PFTBA) before introduction of the sample to ensure that calibration had not shifted. Additionally, a blank of methanol solvent used for extraction was also injected into the machine. Sample injection was performed using a split-less mode using five microliters as the volume. Temperature at the injector was set at 280 °C. The oven temperature was programmed as follows: 70 °C for 2 min and increased at a rate of 25 °C/min to 150 °C and held for 2 min, then increased at a rate of 3 °C/min to 200 °C and held for 2 min, and finally 8 °C/min to 280 °C and held for 10 min. The ionization voltage of MS-analysis was controlled by the EI procedure with the ion source heat of 280 °C. The total GC-MS running time was 45.867 min. The relative proportional percentage of each component was calculated by comparing its average peak area to the total area. The analysis of the mass spectrum of the GC-MS utilized the National Institute of Standard and Technology (NIST) database, which contains over 62,000 patterns. The spectrum of the identified components was matched against the spectrum of known components stored in the NIST library.

## 3. Results

### 3.1. Plant Identification

Plants collected from various parts of Nwoya district (*E. grandis* and *C. citratus*) were first properly pressed in plant presses and morphologically identified. Confirmation of these plant species was performed at Makerere University Botany Herbarium, Kampala, Uganda. The identification results showed that the eucalyptus leaves used were *Eucalyptus grandis* with plant taxonomy as follows: Kingdom: Plantae; Division: Spermatophyta; Subphyllum Angiospermae; Class Dicotyledonae; Order: Myrtales; Family: Myrtaceae; Genus *Eucalyptus*; Species: *grandis* W. Hill ex Maiden; while for the lemon grass leaves as *Cymbopogon citratus*; Kingdom: Plantae; Division: Magnoliophyta; Class: Liliopsida; Subclass: Commelinidae; Order: Poales; Family: Poaceae; Genus: *Cymbopogon*; Species: *citratus* (DC. Ex Nees) Stapf.

### 3.2. Mosquito Confirmation

Confirmation of reared mosquito species at the Uganda Virus Research Institute insectary was conducted using PCR following the protocol of [55]. All mosquito species used were laboratory-reared *An. gambiae* Kisumu strain.

### 3.3. Isolating and Fractionating the Essential Oils

The extracted EOs by LLE were obtained by decanting, weighing, and pooling together. About 5–8 mL of EOs were obtained from 1.68 kg of Eucalyptus leaves and 0.86 kg of lemon grass leaves to obtain about 5.6 and 8.4 mls of Eucalyptus essential oils (EEOs) and lemon grass essential oils (LEOs) respectively. These were preserved at −20 °C until needed for further testing.

### 3.4. Essential Oil Yield

The yields of Eucalyptus essential oils (EEOs) and lemon grass essential oils (LEOs) are shown in Table 1 and Table 2.

### 3.5. GC-MS Analysis

The EEOs and LEOs of the leaves from *E. grandis* and *C. citratus* were identified by GC-MS analysis to establish the chemical compounds present. The percentage composition of each major compound was determined based on peak area normalization in the GC-MS chromatogram. Results were adjusted according to the NIST library. The chromatographs of both oils showed 18 and 19 peaks for *E. grandis* and *C. citratus*, respectively, and these were interpreted as shown in Table 3. Analysis of EEOs showed that the first compounds were L-alpha-terpineol (33.2%) and Eucalyptol (18%), and the rest were less than 10% and for Lemon grass. The LEOs were Lavandulol, methyl ether (47.2%), and Citral (12.96%) as the main compounds (Table 3). The minority compounds presenting less than 10% composition are also shown in Table 3.

### 3.6. Repellency and Spatial Repellent Index

The behavioral responses to essential oil concentrations of EEO and LEO are presented (Table 4 and Table 5). Variations among the different concentration levels of EOs are revealed (Figure 3 and Figure 4). The most notable response in *An. gambiae* was observed at a concentration of 40% EEO (Table 4) while that of LEO was observed at 80% (Table 5), suggesting mosquitoes were more sensitive to the active ingredients in the EEOs than in LEOs. The spatial activity index varies from −1 to 1:0 indicates no response; −1 indicates that all mosquitoes moved into the treatment chamber, resulting in an attractant response, and 1 indicates that all the mosquitoes moved into the control chamber (away from the treatment source), resulting in a spatial repellent response. Both EEOs and LEOs showed both negative and positive levels of spatial repellency with no statistically significant variances *p* ≤ 0.05). Highest spatial activity was observed at 16 for the EEO at a concentration of 40% and 11 for lemon grass at a concentration of 80%. The highest spatial repellency against *An. gambiae* was observed at a concentration of 40% (SAI = −0.39) for EEOs, and 80% (SAI = 0.12) for LEOS, both showing significant differences from the control group (*p* < 0.05). Forty percent and 80% compositions for EEOs and LEOs, respectively, demonstrated the most promising spatial activity index against *An. gambiae*. The effective dose composition of EEO at 50% ED_50_ was 45.49% and at 90, ED_90_ was 51.35%. For *C. citratus*, no LEO ED_50_ detection was observed at 50% and at 90% the ED_90_ was 0.052%.

The spatial activity index of EEO and LEO on *An. gambiae* are shown in Figure 5 and Figure 6, showing varying spatial activity indices with responses varying between the negative and positive values. Both EEOs and LEOs showed both negative and positive levels of spatial repellency with no statistically significant variances (*p* ≤ 0.05). The highest activity was observed at 16 for the EEO at a concentration of 40% and for lemon grass this was 11 at 80%. The highest spatial repellency against *An. gambiae* was observed at a concentration of 40% (SAI = −0.39) for EEOs, and 80% (SAI = 0.12) for LEOS, with both showing significant differences from the control group (*p* < 0.05). Forty percent and 80% compositions for the EEOs and LEOs, respectively, demonstrated the most promising spatial activity index against *An. gambiae*.

## 4. Discussion

For many generations, natural plants and their products have been used to drive away pest insects as well as for human protection from bites [56,57,58,59]. They have been planted around homes for protection from bites of various insects. Plant extracts have been further developed; first, by different means of extraction and by designing products that are applied to the human skin for protection from insect bites [56,57,58,59]. These products obtained from natural plants could either be crude extracts, essential oils [60,61], or further synthesized into useful products [57,62]. Natural repellents through the release of various compounds offer personal protection against many insect bites [1,63].

Our study established the spatial activity and composition of two EOs obtained from the leaves of aromatic plants *E. grandis* and *C. citratus*. A significant increase in the attraction response of *An. gambiae* Kisumu strain to experimental EEOs and LEOs following exposure in the Y-tube olfactometer treatment chamber compared to the controls was similar to what has been observed in some studies [17,19,20,54,57]. More importantly, from this study, the findings have advanced our understanding of the spatial activity of the EEOs and LEOs as well as the chemical compounds present in *E. grandis* and *C. citratus* as aromatic plants. Increase in spatial activity index was observed with an increase in the concentration of both EOs. The activity dosage ranged between 40 and 80%, which indicated that mosquitoes were influenced by the compounds emitted by both oils. Both EOs showed reasonable repellency. A similar repellency for EEO was reported by [58] and for LEO at 60%, which is almost similar to what we observed. The minimum application volume to be effective was slightly different for both oils being slightly lower for the EEO than LEO. These observed responses in mosquito behavior may be due to the differences in the composition of the plants and also to a hyperactive olfactory response of female *Anopheles* mosquitoes elicited by the EO compounds [58]. The findings support a heightened sensory mechanism of action as there was an increase in the number of mosquitoes landing in the treatment portion of the tube overall and fewer mosquitoes in the controls as the concentration of the essential oils increased. Heightened olfactory acuity can result from extensive grooming, especially of the antennae [64,65]. In terms of the effective dosage to be applied for the EEO and LEO, *E. grandis* showed a much higher percentage at ED_50_ and ED_90_ than *C. citratus*. For *C. citratus*, the LEO ED_50_ could not be detected and at ED_90_ it was very low. This may suggest that LEOs might have a slightly higher potency than the EEOs. However, more studies need to be conducted to ascertain these findings because this study did not specifically isolate the different compounds for spatial response activities.

From the few studies presented on spatial activity, variations in the attraction of mosquito species and insects to different repellents and attraction are shown [62,66,67,68]. Essential oil molecules may disrupt the insect’s ability to detect host-seeking chemical signals and reduce bites by binding to the mosquito’s olfactory receptors [66]. There could also be a possibility of the mosquito’s receptors interacting with the essential oil molecules, which could trigger a repellent response in the mosquito, causing it to avoid the treated area or individual [35]. The precise mechanism by which EOs disrupt the mosquito’s olfactory system may vary depending on the specific compounds present in the oil and their interactions with the complex array of receptors involved in host-seeking behavior [66,69,70,71,72].

In a slightly different tone, only a few studies have investigated the volatile oil composition of *E. grandis* and *C. citratus* varieties and some representations from parts of the world include those performed by [70,73]. The fresh leaves produced pale yellow EOs with a yield between 0.7 for EEO and 1.7 for LEO, which is in accordance with what has been reported for EEOs and LEOs in plants from related studies [13,72,74,75]. Our composition is generally consistent with the chemical profiles of the EOs of some of the EEOs and LEOs from related plant species grown in other countries, where a majority of these were found in Indonesia, Turkey, Cyprus, and China [72,76]. Composition of EEOs differed slightly from what was discovered in [37]; however, the species were different and the climate and environmental conditions were likely slightly different from ours.

For the GCMS composition of the EOs, 18 and 19 peaks were obtained after GCMS analysis in EEOs and LEOs. Some of the EOs presented are similar to what has been found in other studies but the percentage composition is very different. Also the composition of these compounds varies from what was found in other studies [70,72,73]. This could be that the plant species are grown in different climatic zones and are presented with different environmental factors. It could be that the species difference is a contributing factor, along with the age of the plant that was harvested, the chemical and physical conditions of the growth environment, as well as the season of harvesting (season, location, climate, soil, and developmental stage) [68,71,72]. All these factors may have played a part in the presentation of different composition of EEOS and LEOS.

Alpha-terpineol in Eucalyptus had the highest composition in the EEOs. It is one of the most popular fragrant ingredients used in household cleaning products, perfumes, cosmetics, and is used to flavor beverages and foods [74,75,76]. It also has several important medicinal and biological properties [70,75,76,77,78,79] Eucalyptol, which was the second highest, is commonly used for manufacturing of cosmetic products and has many health and biological beneficial uses as well [70]. For *C. citratus*, the LEOs, Lavandulol-methyl-ether had the highest composition, though it was not as highly explored as its other components which could also have several important medicinal and biological properties. Citral, which was the second highest in the LEOs, is commonly used for manufacturing of cosmetic products, and has many health and biological beneficial uses as well [77,78,79]. Citral is well-known for its strong, refreshing lemon-like scent and is widely used in perfumery and cosmetics for its aromatic properties [35]. For the rest of the minority compounds found in the EEOs and LEOs, which were not examined, they could present a potential repellent action, which could still have had a significant effect on the spatial behavior of the mosquitoes. For example, 1,8-cineole and limonene could have been present in low composition but might have played a significant role in the spatial activities. Therefore, there was a reduction in mosquito numbers as mosquitoes refused (were repelled) to perch on the treatment arm that had been oiled with EEOs and LEOs due to terpenoids and other class contents that can be toxic for mosquitoes. *Eucalyptus grandis* and *C. citratus* EOs both contained a variety of terpenes and sesquiterpenes. Both EOs showed high levels of composition which was dominated by monoterpenes. For Eucalyptus, eucalyptol, also known as 1,8-Cineole, was very much present though it was not the highest in composition. For *C. citratus*, citral, which is a monoterpene, was present. Other compounds belonging to other groups were in low in composition. The components were not isolated singly so each individual‘s effect was not studied separately. This is an area for further exploration.

## 5. Limitations

This study tested only the activity of aromatic essential oils from plants on one species of mosquito but most probably the product can. Further work may be conducted on other mosquito species. This experiment was conducted on lab-reared mosquitoes that may exhibit different behaviors compared to natural field populations. This needs to be further explored and tested in the semi field experiments and on human subjects. Only two products were tested for the effective dosage and spatial responses; there are many more compounds that could be tested. These may also have spatial repellent chemicals that could affect mosquito behavior differently so further studies are recommended. Specifically, from these results, a broader functionality of volatile repellents beyond their current application (to prevent human biting) could facilitate the development and expanding label uses of available products. It is hoped that this may further incentivize the discovery, development, and evaluation of new spatial repellent products for mosquito and other arthropod-borne disease prevention. In summary, both EEOs and LEOs offer a promising approach for spatial mosquito control, particularly against *An. gambiae* species. Their ability to deter mosquitoes from entering and feeding can significantly reduce the risk of malaria transmission.

## 6. Conclusions

Mosquitoes were repelled by the EEO and LEO test compounds. GCMS analysis showed that the clear yellow oil of EEOs contained 18 compounds, with the highest being L-alpha-terpineol while for the LEOS, a clear yellow color was also presented, containing 19 compounds with the most prevalent compound being Lavandulol-methyl-ether. Both EEOs and LEOs offer a promising approach for spatial mosquito control, particularly against the *An. gambiae* species and could be further developed into useful products to protect humans and animals.

## Figures and Tables

**Figure 1 biology-14-01768-f001:**
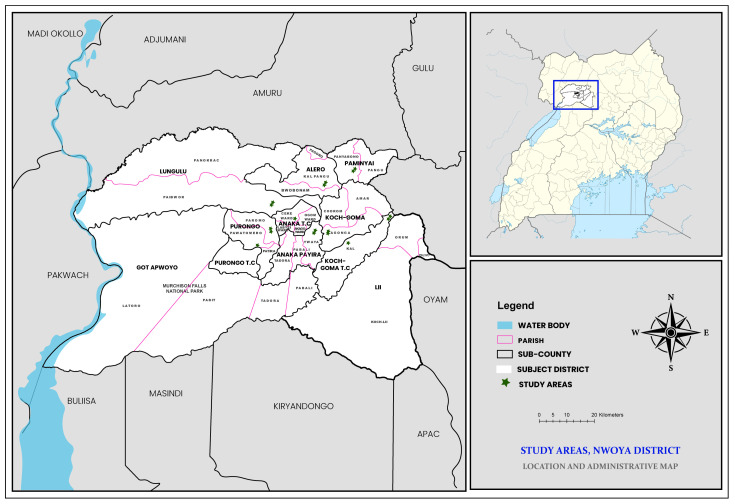
Nwoya district (showing the various sampled sub counties).

**Figure 2 biology-14-01768-f002:**
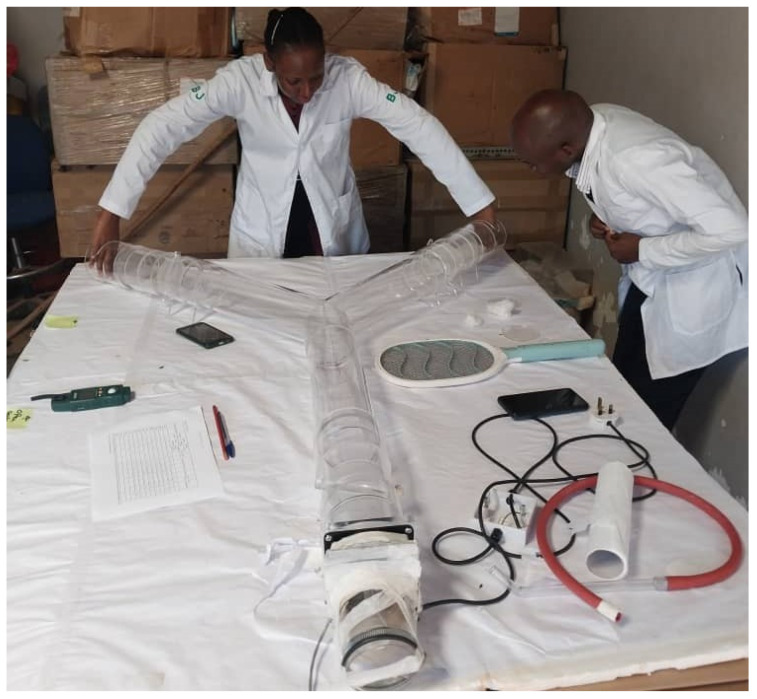
Y-tube olfactometer set up used in the study.

**Figure 3 biology-14-01768-f003:**
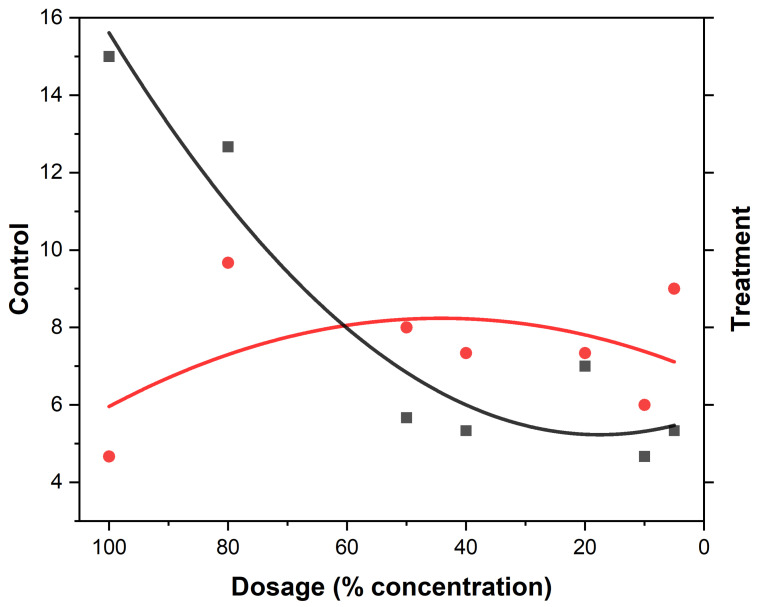
Lemon grass mosquito repellency. The black squares represent the treatment (mosquito responses being repelled) to LEOs and the red dots the control (attraction to the oil). The curves represent the gradient at the different EO compositions.

**Figure 4 biology-14-01768-f004:**
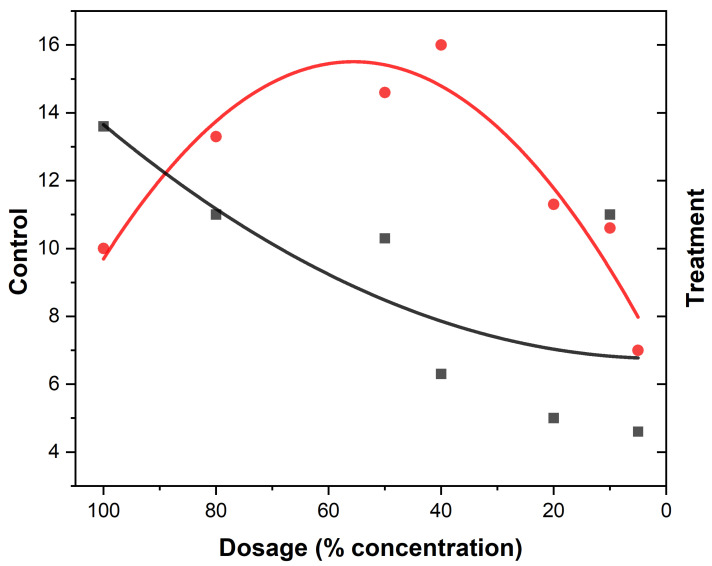
Eucalyptus mosquito repellency. The red dots represent the treatment (mosquito responses being repelled) to EEOs and the black squares the control (attraction to the oil) at different concentrations of the EOs. The red curve indicates the treatment while black curve are the control.

**Figure 5 biology-14-01768-f005:**
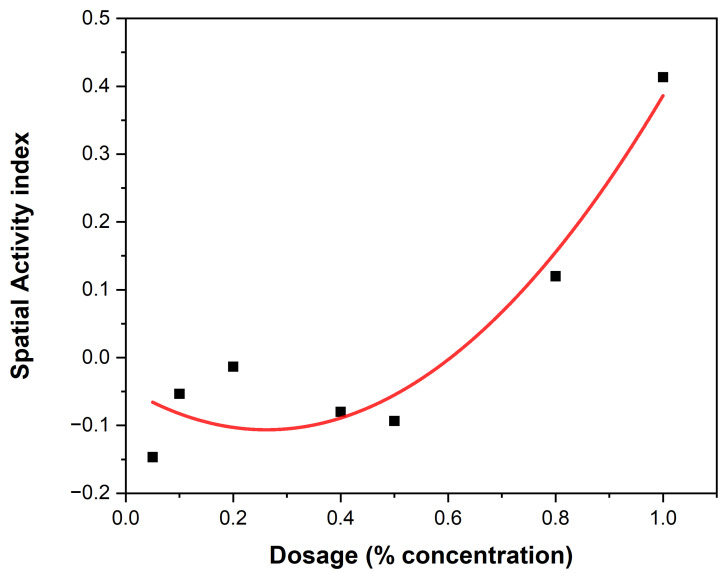
EEOs spatial activity index.

**Figure 6 biology-14-01768-f006:**
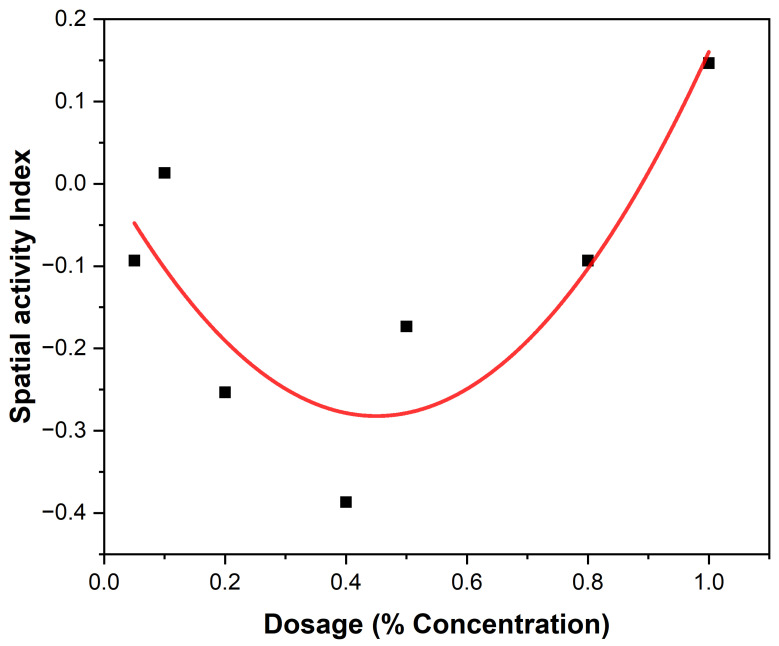
LEOs spatial activity index.

**Table 1 biology-14-01768-t001:** LEOs obtained from *Cymbopogon citratus*; temperature (60–70 °C); 3 h per run.

Run	Weight (g)	Yield (mL)
1	173.5	1.7
2	173.5	1.8
3	173.5	1.7
4	173.5	1.6
5	173.5	1.6

**Table 2 biology-14-01768-t002:** EEOs obtained from *Eucalyptus grandis*, temperature (60–70 °C), 3 h per run.

Run	Weight (g)	Yield (mL)
1	210	0.6
2	210	0.7
3	210	0.7
4	210	0.7
5	210	0.8
6	210	0.7
7	210	0.8
8	210	0.6

**Table 3 biology-14-01768-t003:** Chemical composition of compounds after gas chromatography–mass spectrophotometer (GC-MS) analysis.

Name of Compound	*Eucalyptus grandis*	*Cymbopogon citratus*
	Relative Percentage Amount	
(+)-1-Cyano-d-camphidine	4.53	-
(+)-4-Carene	1.56	-
(+/−)-Lavandulol, methyl ether	-	47.21
(E)-gamma-Atlantone	3.37	-
14-Hydrocycaryophyllene	1.56	0.25
2,2, 5,5-Terthiophene	1.14	-
2,3-Epoxyjuanislamin	1.63	-
2,4-Cholestadien-1-one	1.83	-
4-Hydroxy- beta-ionone	2.41	-
Alpha-Campholenal	6.96	-
Caryophyllene	-	1.94
Caryophyllene oxide	-	3.35
Citral	-	12.63
Ethyl geranate	-	0.18
Eucalyptol	18.88	-
Eugenol	-	5.97
Gamma-Muurolene	-	6.46
Gamma-Terpinene	8.98	-
Geranyl acetate	-	3.12
Geranyl citronellate	-	3.95
Gossonorol	-	0.52
L-alpha- Terpineol	33.21	-
Lanost-8-en-3-ol, (3 beta)-	0.03	-
Levomenol	-	0.15
Linalool	-	5.79
Lithocholic acid	1.32	-
Marrubiin	0.05	-
Methanone	1.38	-
Phytol	5.81	-
Phytyl decanoate	2.2	-
Sesquirosefuran	-	1.97
Tau-muurolol	-	0.36
Trans-geranyl geraniol	-	4.42
Trans-isoeugenol	-	1.03
Trans-sesquisabinene hydrate	-	0.3
Unibelliprenin	-	0.38

**Table 4 biology-14-01768-t004:** Response of female *Anopheles gambiae* to EEOs.

Run (N)	Treatment	Control	Non-Reactants	(Mean Treatment ± SE)	(Mean Control ± SE)
1 (25)	10	15	0	10 ± 1.7	13.7 ± 1.7
2 (25)	11	10	4		
3 (25)	9	16	0		
1 (25)	13	10	2	13.33 ± 0.9	11 ± 0.9
2 (25)	13	12	0		
3 (25)	14	11	0		
1 (25)	11	14	0	14.67 ± 2.2	10.33± 2.2
2 (25)	17	8	0		
3 (25)	16	9	0		
1 (25)	17	3	5	16 ± 3.6	6.33 ± 3.6
2 (25)	19	6	0		
3 (25)	12	10	3		
1 (25)	10	9	6	11.33 ± 2.8	5 ± 2.8
2 (25)	16	4	5		
3 (25)	8	2	15		
1 (25)	10	12	3	10.7 ± 0.8	11 ± 0.8
2 (25)	10	12	3		
3 (25)	12	9	4		
1 (25)	7	5	13	7 ± 1.2	4.67 ± 1.2
2 (25)	9	3	13		
3 (25)	5	6	14		

N = total number of mosquitoes tested, SE = standard error.

**Table 5 biology-14-01768-t005:** Response of female *Anopheles gambiae* to LEOs.

Dosage	Run (N)	Treatment	Control	Non-Reactants	(Mean Treatment ± SE)	(Mean Control ± SE)
100%	1	10	5	10	15 ± 4.52	4.67 ± 4.52
100%	2	11	8	6		
100%	3	24	1	0		
80%	1	13	4	8	12.67 ± 2.41	9.67 ± 2.41
80%	2	15	10	0		
80%	3	10	15	0		
50%	1	6	7	11	5.67 ± 1.12	8 ± 1.12
50%	2	4	10	10		
50%	3	7	7	9		
40%	1	6	10	5	5.33 ± 1.6	7.33 ± 1.6
40%	2	8	5	8		
40%	3	2	7	15		
20%	1	5	5	14	7 ± 1.4	7.3 ± 1.4
20%	2	9	6	8		
20%	3	7	11	7		
10%	1	3	6	15	4.7 ± 1.2	6 ± 1.2
10%	2	8	5	15		
10%	3	3	7	15		
5%	1	4	8	13	5.3 ± 1.3	9 ± 1.3
5%	2	6	10	6		
5%	3	6	9	7		

N = total number of mosquitoes tested, SE = standard error.

## Data Availability

All data have been included in the submitted manuscript.

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
