# Peer review of "Spatial Exposure Responses of Malaria Vectors to Eucalyptus grandis (W. Hill ex Maiden) and Cymbopogon citratus (DC.) Stapf Essential Oils"

_biology, 2025, doi:10.3390/biology14121768_

Round 1
Reviewer 1 Report
Comments and Suggestions for Authors
The manuscript addresses an important problem in malaria control: the search for effective plant-based repellents in the face of increasing insecticide resistance. There is some points needed to be clearer as follows:
A clear schematic or image of the Y-tube olfactometer setup should be included in the Methods section to improve reader understanding.
Please clarify what is the “treated surface.”
What exactly was placed in the control chamber? This needs to be specified.
Line 194: The text refers to “methanolic extract,” but the Methods describe steam distillation. Please clarify this inconsistency.
The manuscript inconsistently uses “Lemon grass” and “Cymbopogon citratus.” Only one term should be used consistently throughout.
Figures 4 and 5 may not be necessary, since the chemical components are already presented in Table 3.
The manuscript refers to “Figure 8,” but no such figure is included. Please correct this.
Author Response
Reviewer 1 Open Review (x) I would not like to sign my review report
( ) I would like to sign my review report Quality of English Language ( ) The English could be improved to more clearly express the research.
(x) The English is fine and does not require any improvement.
| Yes | Can be improved | Must be improved | Not applicable | |
| Does the introduction provide sufficient background and include all relevant references? | ( ) | (x) | ( ) | ( ) |
| Is the research design appropriate? | ( ) | (x) | ( ) | ( ) |
| Are the methods adequately described? | ( ) | (x) | ( ) | ( ) |
| Are the results clearly presented? | ( ) | (x) | ( ) | ( ) |
| Are the conclusions supported by the results? | ( ) | (x) | ( ) | ( ) |
| Are all figures and tables clear and well-presented? | ( ) | (x) | ( ) | ( ) |
Comments and Suggestions for Authors
The manuscript addresses an important problem in malaria control: the search for effective plant-based repellents in the face of increasing insecticide resistance. There is some points needed to be clearer as follows:
A clear schematic or image of the Y-tube olfactometer setup should be included in the Methods section to improve reader understanding.
Please clarify what is the “treated surface.”
What exactly was placed in the control chamber? This needs to be specified.
Line 194: The text refers to “methanolic extract,” but the Methods describe steam distillation. Please clarify this inconsistency.
The manuscript inconsistently uses “Lemon grass” and “Cymbopogon citratus.” Only one term should be used consistently throughout.
Figures 4 and 5 may not be necessary, since the chemical components are already presented in Table 3.
The manuscript refers to “Figure 8,” but no such figure is included. Please correct this.
Author Response:
Comment 1: A clear schematic or image of the Y-tube olfactometer setup should be included in the Methods section to improve reader understanding.
Respone 1: The Y tube olfactometer set up has been included in the manuscript.
Comment 2: Please clarify what is the “treated surface.”
Response 2: Treated surface refers to the surface where the tested dosages (EEO and LEO) of the test compounds of the EOs were placed to release their compounds.
Comment 3: What exactly was placed in the control chamber? This needs to be specified.
Response 3: In the control chamber, pure olive oil was placed.
Comment 4: Line 194: The text refers to “methanolic extract,” but the Methods describe steam distillation. Please clarify this inconsistency.
Response 4: The inconsistency in line 194 has been rectified to remove methanolic extracts.
Comment 5: The manuscript inconsistently uses “Lemon grass” and “Cymbopogon citratus.” Only one term should be used consistently throughout.
Response 5: Only one term is used consistently in the manuscript and this is Cymbopogon citratus.
Comment 6: Figures 4 and 5 may not be necessary, since the chemical components are already presented in Table 3.
Response 6: Figure 4 and 5 have been removed out of manuscript and only the results are presented in Table 3.
Comment 7: The manuscript refers to “Figure 8,” but no such figure is included. Please correct this.
Response 7: This has been corrected and the figures are now properly numbered. Overall, the English has been edited.
Reviewer 2 Report
Comments and Suggestions for Authors
Comments to the Author
-
The order of writing is inconsistent. Please carefully review the use of scientific names, abbreviations, and other technical terms to ensure consistency throughout the manuscript.
-
The study includes only three replicates. Please justify whether this number is sufficient for statistical reliability.
-
Statistical data from the Probit analysis should be presented, including EC values, to strengthen the results.
-
Both essential oils examined in this study have been widely investigated in previous research. The introduction should include a summary of prior studies and clearly identify the research gap addressed by this work.
-
The GC–MS graph is not clearly presented. Please improve the figure quality to ensure readability and accuracy.

Comments on the Quality of English Language
The English should be written in a scholarly manner and consistent across sentences.
Author Response
Open Review (x) I would not like to sign my review report
( ) I would like to sign my review report Quality of English Language (x) The English could be improved to more clearly express the research.
( ) The English is fine and does not require any improvement.
| Yes | Can be improved | Must be improved | Not applicable | |
| Does the introduction provide sufficient background and include all relevant references? | ( ) | ( ) | (x) | ( ) |
| Is the research design appropriate? | ( ) | (x) | ( ) | ( ) |
| Are the methods adequately described? | ( ) | (x) | ( ) | ( ) |
| Are the results clearly presented? | ( ) | ( ) | (x) | ( ) |
| Are the conclusions supported by the results? | (x) | ( ) | ( ) | ( ) |
| Are all figures and tables clear and well-presented? | ( ) | ( ) | (x) | ( ) |
Comments and Suggestions for Authors
Comments to the Author
-
The order of writing is inconsistent. Please carefully review the use of scientific names, abbreviations, and other technical terms to ensure consistency throughout the manuscript.
-
The study includes only three replicates. Please justify whether this number is sufficient for statistical reliability.
-
Statistical data from the Probit analysis should be presented, including EC values, to strengthen the results.
-
Both essential oils examined in this study have been widely investigated in previous research. The introduction should include a summary of prior studies and clearly identify the research gap addressed by this work.
-
The GC–MS graph is not clearly presented. Please improve the figure quality to ensure readability and accuracy.
Author Response:
COMMENT ONE
The order of writing is inconsistent. Please carefully review the use of scientific names, abbreviations, and other technical terms to ensure consistency throughout the manuscript.
RESPONSE ONE
Thank you reviewer for this comment. We have adjusted the this. The writing over the whole text has been greatly revised after careful checking of the scientific names abbreviations to ensure consistency. From line 59 through the entire manuscript this has been revised.
COMMENT TWO
The study includes only three replicates. Please justify whether this number is sufficient for statistical reliability.
RESPONSE TWO
Thank you reviewer for this response. We agree we could have used more replicates to allow for variance of the study. Three replicates were used as a minimum because the mosquitoes that were used were reared and could only be available up to a certain number. Only a limited number of mosquitoes could be reared for the various testing’s. We allowed for this due to production and time constraints.
COMMENT THREE
Statistical data from the Probit analysis should be presented, including EC values, to strengthen the results.
RESPONSE THREE
Thank you reviewer for this comment. The probit analyses case values in particular ED 50 and 90 have been added to the results section (Line 362 to 264) to improve the results.
COMMENT FOUR
Both essential oils examined in this study have been widely investigated in previous research. The introduction should include a summary of prior studies and clearly identify the research gap addressed by this work
RESPONSE FOUR
There has a been a great improvement in the writing and many other studies have been added to the introduction and overall manuscirpt. The whole manuscript has been revised to take care of the writing and references.
COMMENT FIVE
The GC–MS graph is not clearly presented. Please improve the figure quality to ensure.
RESPONSE FIVE
The GC-MS graph has been removed.
Reviewer 3 Report
Comments and Suggestions for Authors
The manuscript entitled “Spatial exposure responses of malaria vectors to Eucalyptus grandis (W. Hill ex. Maiden) and Cymbopogon citratus (DC.) Stapf. essential oils.” represents an interesting investigation into a topic of extreme importance, especially in the areas studied by the authors.
There is considerable scientific rigor in the structure of the manuscript, the organization of the experimental procedures, and the analysis of the data collected.
I believe it is important to highlight just a few formal aspects that are nevertheless significant for the manuscript itself:
- In line 103, replace canun with canum, the correct name of the basil species cited by the authors
- Correct the genus and species names of plant species throughout the text; these should always be in italics, which is often not the case
- The chromatograms shown in Figures 2 and 3 are difficult to read; this is probably due to image definition issues. If it were possible to make them clearer, it would be very helpful for understanding purposes; if this is impossible due to editorial limitations, the authors could consider introducing more usable images as supplementary material.
One last but no less important consideration: regardless of the logical considerations regarding the most representative compounds found in the GC-MS analyses, I would not rule out a potential repellent action of the minority compounds, which could still have had a significant effect in this regard thanks to their synergistic action. Even without necessarily engaging in excessive speculation, the authors could argue this point in the discussion section by referring to the available scientific literature on the subject.
Author Response
Open Review (x) I would not like to sign my review report
( ) I would like to sign my review report Quality of English Language ( ) The English could be improved to more clearly express the research.
(x) The English is fine and does not require any improvement.
| Yes | Can be improved | Must be improved | Not applicable | |
| Does the introduction provide sufficient background and include all relevant references? | (x) | ( ) | ( ) | ( ) |
| Is the research design appropriate? | (x) | ( ) | ( ) | ( ) |
| Are the methods adequately described? | (x) | ( ) | ( ) | ( ) |
| Are the results clearly presented? | ( ) | (x) | ( ) | ( ) |
| Are the conclusions supported by the results? | (x) | ( ) | ( ) | ( ) |
| Are all figures and tables clear and well-presented? | ( ) | (x) | ( ) | ( ) |
Comments and Suggestions for Authors
The manuscript entitled “Spatial exposure responses of malaria vectors to Eucalyptus grandis (W. Hill ex. Maiden) and Cymbopogon citratus (DC.) Stapf. essential oils.” represents an interesting investigation into a topic of extreme importance, especially in the areas studied by the authors.
There is considerable scientific rigor in the structure of the manuscript, the organization of the experimental procedures, and the analysis of the data collected.
I believe it is important to highlight just a few formal aspects that are nevertheless significant for the manuscript itself:
- In line 103, replace canun with canum, the correct name of the basil species cited by the authors
- Correct the genus and species names of plant species throughout the text; these should always be in italics, which is often not the case
- The chromatograms shown in Figures 2 and 3 are difficult to read; this is probably due to image definition issues. If it were possible to make them clearer, it would be very helpful for understanding purposes; if this is impossible due to editorial limitations, the authors could consider introducing more usable images as supplementary material.
One last but no less important consideration: regardless of the logical considerations regarding the most representative compounds found in the GC-MS analyses, I would not rule out a potential repellent action of the minority compounds, which could still have had a significant effect in this regard thanks to their synergistic action. Even without necessarily engaging in excessive speculation, the authors could argue this point in the discussion section by referring to the available scientific literature on the subject.
Author Response:
COMMENT ONE
In line 103, replace canun with canum, the correct name of the basil species cited by the authors
RESPONSE ONE
Thank you reviewer for this comment. The spelling has been edited and it is now found on Page 1, Line 115.
COMMENT TWO
Correct the genus and species names of plant species throughout the text; these should always be in italics, which is often not the case
RESPONSE TWO
Thank you for this comment. Names of the genera and plant species have been revised through the entire manuscript.
COMMENT THREE
The chromatograms shown in Figures 2 and 3 are difficult to read; this is probably due to image definition issues. If it were possible to make them clearer, it would be very helpful for understanding purposes; if this is impossible due to editorial limitations, the authors could consider introducing more usable images as supplementary material.
RESPONSE THREE
We thank the reviewer for this comment. It might not be possible to adjust the figures of the chromatogram so we have decided to leave them out of the manuscript.
COMMENT FOUR
One last but no less important consideration: regardless of the logical considerations regarding the most representative compounds found in the GC-MS analyses, I would not rule out a potential repellent action of the minority compounds, which could still have had a significant effect in this regard thanks to their synergistic action. Even without necessarily engaging in excessive speculation, the authors could argue this point in the discussion section by referring to the available scientific literature on the subject.
RESPONSE FOUR
We have included a small section on the minority compounds obtained from the GCMS analysis in the results and discussion on page 12-13 and also on Page 20, Line 468 to 473.
Round 2
Reviewer 2 Report
Comments and Suggestions for Authors
Your writing has improved. Please double-check all abbreviations, full terms, scientific names, and the use of italics. I have highlighted some examples for your attention.

Author Response
Open Review (x) I would not like to sign my review report
( ) I would like to sign my review report Quality of English Language ( ) The English could be improved to more clearly express the research.
(x) The English is fine and does not require any improvement.
| Yes | Can be improved | Must be improved | Not applicable | |
| Does the introduction provide sufficient background and include all relevant references? | (x) | ( ) | ( ) | ( ) |
| Is the research design appropriate? | (x) | ( ) | ( ) | ( ) |
| Are the methods adequately described? | (x) | ( ) | ( ) | ( ) |
| Are the results clearly presented? | (x) | ( ) | ( ) | ( ) |
| Are the conclusions supported by the results? | (x) | ( ) | ( ) | ( ) |
| Are all figures and tables clear and well-presented? | (x) | ( ) | ( ) | ( ) |
Response 1 (General comments) The whole manuscript has been properly checked for any inconsistencies and these have been rectified. Comments and Suggestions for Authors
Your writing has improved. Please double-check all abbreviations, full terms, scientific names, and the use of italics. I have highlighted some examples for your attention.
Response 2
All abbreviations have repeatedly been verified, cross checking the scientific names and scientific nomenclature properly rectified.